# Disaster Risk Mapping: A Desk Review of Global Best Practices and Evidence for South Asia

**Giriraj Amarnath \*** , **Upali A. Amarasinghe and Niranga Alahacoon**

International Water Management Institute (IWMI), 127 Sunil Mawatha, Battaramulla, Colombo 10120, Sri Lanka; u.amarasinghe@cgiar.org (U.A.A.); n.alahacoon@cgiar.org (N.A.)
\* Correspondence: a.giriraj@cgiar.org; Tel.: +94-11-2880000

**Abstract:** The frequency, intensity, and variability of natural hazards are increasing with climate change. Detailed sub-national information on disaster risks associated with individual and multi-hazards enables better spatial targeting of adaptation and mitigation measures. This paper reviews the global best practices of disaster risk mapping (DRM) to assess the nature and magnitude of disasters, and the vulnerability and risks at the sub-national level in South Asian countries. While some global DRMs focus on vulnerability, others assess risks. Most DRMs focus on national-level vulnerability and risks. Those which focus on the sub-national risks have a limited scope and different methodologies for evaluating risks, mainly in relation to the population. Climate change exposes not only people but also many infrastructures, assets and their impacts to disaster risk. For DRMs to be useful tools for sub-national planning, they require a coherent methodology and a high-resolution spatial focus. The vulnerability and risk assessments should focus on different aspects, including population, infrastructure, and assets in various economic sectors of agriculture, industry, and services.

**Keywords:** disaster risk mapping; climate hazards; vulnerability; agriculture; South Asia

## 1. Introduction

Population growth and rapid urbanization are driving the increase in disaster risks. The Bank's Aftershocks report explains that these trends could put 1.3 billion people and USD 158 trillion in assets at risk from river and coastal floods alone [1]. Disaster risks are increasing the world over [2], with major concerns being the increasing frequency, intensity, spatial variability, and unpredictability of natural hazards and their impacts [3]. In South Asia (SA), the incidence of natural hazards increased from about 14 per year in the 1970s to 24, 36, and 47 per year in the 1980s, 1990s, and 2000s, respectively [4], including losses and damages due to natural disasters. Consequently, the number of people affected, the loss of lives, and the costs to economies associated with disasters are also increasing rapidly. The economic losses triggered by natural hazards at present are over USD four billion per year in SA [5].

With rapidly changing demographics, growing economies, and interactions between different sectors, more people, businesses, infrastructure, assets, and economic activities are exposed to disaster risks from natural hazards [6,7]. Often, the most affected people are disadvantaged groups, including women, children, the elderly, disabled people, displaced populations, and religious and ethnic minorities [8,9]. The small- and medium-scale enterprises are among the hardest-hit businesses. Furthermore, the losses to ecosystems and their subsequent impacts often go unnoticed. Therefore, detailed risk assessments are essential to safeguard investments in all sectors and value chains of economic activities [10–15].

The South Asian countries, with 1.5 billion people, have one of the largest populations exposed to natural hazards [16]. A majority of the people in SA still live in rural areas and depend upon agriculture for their livelihoods [17]. The industrial and service sectors, mostly concentrated in urban areas, increasingly contribute to a substantial part of the

gross domestic product. Indeed, all economic sectors face high risks with increasing natural hazards [18]. Understanding the nature and magnitudes of risks of disaster associated with natural hazards is critical to enhancing resilience to the impacts of climate change and accelerating socio-economic development [19,20].

Flood and droughts are recurrent and simultaneously occur in different parts in small to large countries. In SA, the northern and southern parts of India, the Indian state of Bihar, or the small islands such as Sri Lanka are examples [21–24]. Floods create havoc in some regions, while droughts destroy agricultural production and livelihoods within the other areas. Extreme rainfall, temperature, cold- or heatwaves, storms, and cyclones can exacerbate the situation [25].

Investigations of the risks of multiple hazards are also urgently needed since the occurrence of some hazards can exacerbate or even reduce the probability of secondary hazards [26]. Therefore, a risk assessment should account for the potential interactions between multiple hazards, such as precipitation with landslides or fires with drought or earthquakes, etc. [27]. Moreover, risk assessments and management must account for the upstream and downstream effects in river basins. High amounts of rainfall upstream can create flooding downstream [28]. Conversely, substantial water storage upstream to mitigate climate change impacts can contribute to droughts downstream [29]. Such up- and down-stream interactions are especially true in South Asia due to the transboundary nature of many large river basins.

Transboundary river basins, such as the Indus, Ganges, and Brahmaputra cover a large part of SA. These river basins are already under extreme pressure, on the one hand, due to recurrent natural hazards especially floods and droughts and on the other hand due to the over-exploitation of water resources, especially for agriculture [16,30,31]. The other big river basins such as Krishna, Godavari, Mahanadi, Cauvery, and Narmada in peninsular India that flow through several states also have similar issues with disaster risks. Increasing disaster risks only aggravate the conflicts of water sharing and allocation and constrain regional economic growth in transboundary river basins.

Disaster risk mapping (DRM) is gaining substantial attention lately due to climate change [32]. Yet, many disaster risk management activities do not use sub-national risk maps due to a lack of sufficiently high-resolution information [33]. Historical data estimate and/or model the frequency of occurrence of hazards; however, high-resolution maps tend to reduce the disaster risk from natural hazards by overlooking exposure, vulnerability, and coping capacity. The profiling of disaster risks requires detailed information on the causes, losses, damages, and coping capacity of people and infrastructure [34]. Yet, because of the considerable spatial variation of incidence and frequency of disasters, the sub-national DRMs are critical for regions such as SA [35,36].

The main purpose of this study is to review the best practices available globally for disaster risk mapping (DRM) including the methods and tools for (sub) national risk as-assessment in South Asian countries. The outline of the DRM review in this paper follows Section 2 of the paper, and reviews existing practices in disaster risk mapping. The next section illustrates various approaches to single and multiple hazard maps as well as exposure and vulnerability/capacity maps. Finally, the paper concludes with recommendations for future DRMs.

*Review of Existing Disaster Risk Mapping Tools*

The report only considers open-source and publicly available DRM tools and assessments (Table 1). The geographical focus of these tools varies from most countries in the world to a few countries in a region. The geographic scale of assessment ranges from global and national to the sub-national level. The tools have played an essential role in improving the risk mapping methodology and/or providing geo-spatial information to governments, development organizations, or disaster risk management practitioners.

**Table 1.** Introduction to the analyzed disaster risk tools and assessments.

| No. | Name and Abbreviation (If Commonly Used) | Agency and Year (Updates If Available) | Publicly Accessible Tool | Publicly Available Assessment | Geographical Focus | Geographical Scale of Analysis |
|---|---|---|---|---|---|---|
| 1 | Natural Disaster Hotspot: A Global Risk Analysis (Hotspot s Study) | WB 2005 [37] | No | Yes | The world | Global to sub-national levels |
| 2 | Open Data for Resilience Initiative (Open DRI) | WB/GFDRR 2011 (Continuous updates by countries) [38] | No | Yes | All countries | Global to sub-national levels |
| 3 | Global Risk Data Platform (GRDP) | UNEP/UNISDR 2013 [39] | Yes | No | The world | Global to sub-national levels |
| 4 | Child-centered Risk Assessment: Regional Synthesis of UNICEF Assessments in Asia | UNICEF 2014 (continuous updates by country offices) [40] | No | Yes | Six countries in the Asia-Pacific including Nepal and India in South Asia | National to sub-national levels |
| 5 | South Asia Women's Resilience Index (WRI) | Action Aid 2014 [41] | No | Yes | Countries in South Asia and Japan | National level only |
| 6 | Index for Risk Management (INFORM) | IASC/EC 2015 (updated every half year, last updates from mid-2018) [42] | No | Yes | All countries | Global to the national level (and sub-national levels for individual countries outside South Asia) |
| 7 | The World Risk Index (WRI) (For more information see: http://www.uni-stuttgart.de/ireus/Internationales/WorldRiskIndex/ (accessed on 11 June 2021)) | University of Stuttgart 2015 (updated annually) [43] | No | Yes | All countries | Global to national levels |
| 8 | The GAR Atlas: Unveiling Global Disaster Risk (GAR Atlas) | UNISDR 2017 (updated biennially) [15] | No | Yes | The world | Global to sub-national levels |
| 9 | Atlas of the Human Planet: Global Exposure to Natural Hazards | European Commission 2017 [44] | No | Yes | All countries | Global to national levels |
| 10 | Mapping Multiple Climate-related Hazards in South Asia | IWMI 2017 [16] | No | Yes | South Asia only | Regional to sub-national levels |

Table A1 provides the list of various disaster risk mapping tools and their application. The synthesis following this section compares and contrasts the methodologies of risk assessment, spatial scope, results, and the inclusion of social development in disaster risk mapping.

Conventionally, hazards take two forms: natural or human-induced hazards, although this distinction is increasingly becoming blurred due to the impacts of human activities, e.g., climate change and fracking. Most DRMs reviewed below have focused on natural hazards (Table 2). Only a few have considered risks induced by climate change such as the sea-level rise and by humans such as conflicts, industrial accidents, etc.

**Table 2.** Types of hazards and analyzed tools and assessments.

| No. | Name and Abbreviation | Geo-Physical Hazards | | | | Hydro-Meteorological Hazards | | | | | | Climate Change | Human-Made Hazards |
|---|---|---|---|---|---|---|---|---|---|---|---|---|---|
| | | Volcanos | Earthquakes | Tsunamis | Landslides (Seismic) | Floods | Cyclones | Landslides (Precipitation) | Drought | Heat Waves and Cold Spells/Wind | Wildfires | Sea Level Rise/Storm Surge | Conflict | Industrial Accidents |
| 1 | WB 2005 | Y | Y | Y | N | Y | Y | Y | Y | N | Y | N | N | N |
| 2 | WB/GFDRR 2011 | Y | Y | Y | Y | Y | Y | Y | Y | Y | Y | N | N | N |
| 3 | UNEP/UNISDR 2013 | Y | Y | Y | N | Y | Y | N | N | N | N | N | N | N |
| 4 | UNICEF 2014 | Y | Y | Y | Y | Y | Y | Y | Y | N | N | N | N | N |
| 5 | Action Aid 2014 | N | N | N | N | N | N | N | N | N | N | N | N | N |
| 6 | IASC/EC 2015 | N | Y | Y | N | Y | Y | N | Y | N | N | N | Y | N |
| 7 | University of Stuttgart 2015 | N | Y | N | N | Y | Y | N | Y | N | N | Y | N | N |
| 8 | UNISDR 2017 | Y | Y | Y | N | Y | Y | N | N | Y | N | Y | N | N |
| 9 | EC 2017 | Y | Y | Y | N | Y | Y | N | N | N | N | Y | N | N |
| 10 | IWMI 2017 | N | N | Y | N | Y | Y | N | Y | Y | N | N | N | N |

Note: Yes and No referred as "Y" and "N".

The natural hazards in this review include:

- Geophysical hazards such as earthquakes, tsunamis, volcanoes, and landslides (seismic induced);
- Hydro-meteorological hazards such as cyclones, floods, droughts, landslides (caused by precipitation), extreme rainfall events or heatwaves, and wildfires.

## 2. Materials and Methods

*The Methodology of Risk Assessment*

The core of disaster risk assessments has three components: hazard (H), exposure (E), and vulnerability (V). The hazard component focuses on the probability of occurrence, intensity, and spatial coverage of hazards (Schneiderbauer 2004). Exposure is hazard dependent. It combines the likelihood of an event and the exposure to the danger of various elements and assets such as population, buildings, economic value, GDP, etc. Vulnerability includes both physical and socio-economic vulnerability and varies from hazards to aspects of exposure. The disaster risk is generally a function of the three components H, E, and V [45].

However, the approaches to risk assessments vary. Some use an entirely probabilistic approach, which includes historical data to assess the likelihood of hazards with different return periods, and the vulnerability to disasters with vulnerability curves [46]. The vulnerability curves show the probability of the exceedance of losses of various magnitudes. Others use both probabilistic and deterministic approaches for risk assessments. Here the exposure and vulnerability are based on a composite index of various demographic, social, economic, and environmental factors (IASC/EC 2015; University of Stuttgart 2015; IWMI 2017).

The geometric mean of H, E, and V is the risk function in some assessments, while others use the arithmetic mean. The geometric mean gives low prominence to low values of risk of E or V (and in cases with adaptive capacity). The arithmetic means have equal prominence to each component.

The review in this paper contrasts and compares various types of DRM methodologies (Table 3). The risk assessments in the WB 2005, UNEP/UNISDR 2013, UNISDR 2017, and EC 2017 were entirely probabilistic. IASC/EC 2015 and University of Stuttgart 2015 used probabilistic approaches to assess the exposure to hazards and a deterministic approach to assess vulnerability and adaptive capacity. IWMI used a deterministic approach to determine exposure and susceptibility. UNICEF 2014 focused on only child-centered vulnerability in a few Asian countries. Action Aid 2014 assessed the capacity of disaster-risk management with a specific focus on women's needs using various indicators of economic, infrastructure, social, and institutions of the countries. While most DRMs have conducted multi-hazard analyses, only a few have focused on the impacts of climate change [47]. A majority of the DRMs have assessed the exposure of people and assets to disaster risks.

**Table 3.** Disaster risk components in the analyzed tools and assessments.

| No. | Organization and Year | Geo-Physical Hazards | Hydro-Meteorological Hazards | Probabilistic or Deterministic | Multi-hazard Analysis | Climate Change | Exposure to Assets | Exposure to People | Gender and Age | Vulnerability | Social Development | Capacity |
|---|---|---|---|---|---|---|---|---|---|---|---|---|
| 1 | WB 2005 | Y | Y | Prob | Y | N | Y | Y | N | Y | N | N |
| 2 | WB/GFDRR 2011 | Y | Y | Det | Y | Y | Y | Y | Y | Y | Y | Y |
| 3 | UNEP/UNISDR 2013 | Y | Y | Prob | Y | N | Y | Y | N | Y | N | N |
| 4 | UNICEF 2014 | Y | Y | Both | Y | Y | Y | Y | Y | Y | Y | Y |
| 5 | Action Aid 2014 | N | N | Det | N | N | N | Y | Y | Y | Y | Y |
| 6 | IASC/EC 2015 | Y | Y | Both | Y | N | Y | Y | N | Y | Y | Y |
| 7 | University of Stuttgart 2015 | Y | Y | Both | Y | N | Y | Y | N | Y | Y | Y |
| 8 | UNISDR 2017 | Y | Y | Prob | Y | Y | Y | N | N | Y | N | N |
| 9 | EC 2017 | Y | Y | Prob | N | N | Y | Y | N | N | N | N |
| 10 | IWMI 2017 | Y | Y | Det | Y | Y | Y | Y | N | Y | N | N |

Notes: Yes and No referred as "Y" and"N", Prob—Probabilistics, Det—Deterministic.

First, the hazard information of population grids is calculated using the area-weighted sum of hazard information of grids that lie entirely or partially in the population grid. The product of the estimated hazard value and the total population is the exposed population to hazard. The assessment includes collecting data for developing hazards exposure, vulnerability profiles, and estimating risk at the sub-national level. This initiative supports 31 countries, with 4 in South Asia, including Bangladesh, Nepal, Pakistan, and Sri Lanka at present for sharing, collecting, or/and processing data for different components.

- In South Asia, OpenDRI supports the uploading of hazards and exposure data collected from various government departments and other sources onto open data sharing disaster-risk-information platform.

- At present, the risk platforms are available at http://geodash.gov.bd/ (accessed on 20 April 2021) in Bangladesh, http://drm.moha.gov.np/ (accessed on 1 March 2021) in Nepal, www.disasterinfo.gov.pk (accessed on 20 April 2021) in Pakistan, and http://riskinfo.lk/ (accessed on 20 April 2021) in Sri Lanka.
- Pakistan, Nepal, and Sri Lanka use open access Geonode (GeoNode, which is a web-based platform for developing geospatial information systems. It facilitates the uploading of spatial data and infrastructure http://geonode.org/ (accessed on 20 April 2021), while Bangladesh has created its own information platform.
- OpenDRI supports data collection by mapping buildings and roadways on crowd-sourced OpenStreetMap database. So far, it has mapped 8500 buildings, 93 km of roads, and 50 km of drainage canals in Bangladesh; 2250 schools and 350 health facilities in Nepal; and 130,564 buildings and more than 1000 km of road in Sri Lanka. In Pakistan, it has trained people to use open access Geonode and OpenStreetMap.
- In Sri Lanka, assessments of the impacts of recent floods and the required risk mitigation response in the Gampaha district in the Western province used OpenStreetMap.
- UNEP/UNISDR 2013, UNISDR 2017, and EC 2017 estimate sub-national risk maps using an entirely probabilistic approach. The latter two mainly use the data generated in UNEP/UNISDR 2013.
- IWMI 2017 is a regional assessment and focused on climate-related hazards at the sub-national level in South Asia. It combined the disaster exposure maps estimated at the grid level and the human development index (HDI) available at the district level to assess sub-national level vulnerability in SA. The HDI [48] is based mainly on socio-economic indicators of education, gross national income, and life expectancy at birth. The accuracy of this estimate depends on how far HDI accurately represents the vulnerability to hazards at the sub-national level.
- UNICEF 2014 and Action Aid 2014 focus on disaster risk management of different population segments, in particular women and children. They only have a regional focus—the former on Asia and the Pacific and the latter on the South Asian countries.

## 3. Results

### 3.1. Individual Hazard Maps

Hazards maps show the frequency (number of occurrences), intensity (return periods), or actual exposure of the area to different hazards (Table 4).

- The national-level hazard maps (IASC/EC 2015, University of Stuttgart 2015) are sufficient for comparisons across countries. The primary purpose of these maps is for the use of donors and funding agencies to prioritize support to states for disaster risk management.
- However, the higher resolution hazard maps (UNEP/UNISDR 2013, UNISDR 2017, EC 2017) show considerable details of hazard exposure by combing the frequency and intensity of hazards in a probabilistic framework. They provide useful information for intensive hazards (Figure 1).
- IWMI 2017 exposure maps show the actual exposure regardless of the intensity of hazards. They use remote sensing images to identify the exposed pixels to disasters. The sub-national estimates derived from these are essential for local-level disaster-risk management planning.
- While most risk maps use similar hazard data, the disaster maps, assessed either with probabilistic or deterministic methods, generate different risk profiles for some regions. IASC/EC 2015 shows an extensive earthquake profile for India, but the UNEP/UNISDR 2013 and UNISDR 2017 display large parts which are free of seismic hazards.
- The hazard maps with more extended return periods show the exposure due to intensive hazards (i.e., frequent or short return periods with low intensity). However, with monsoonal and El Niño/Southern Oscillation (ENSO) dominated weather patterns, many locations in South Asia are exposed to recurrent or shorter (4–6 years) return

period floods and droughts. The aggregate losses are generally more substantial due to extensive rather than intensive hazards (UNISDR 2015).

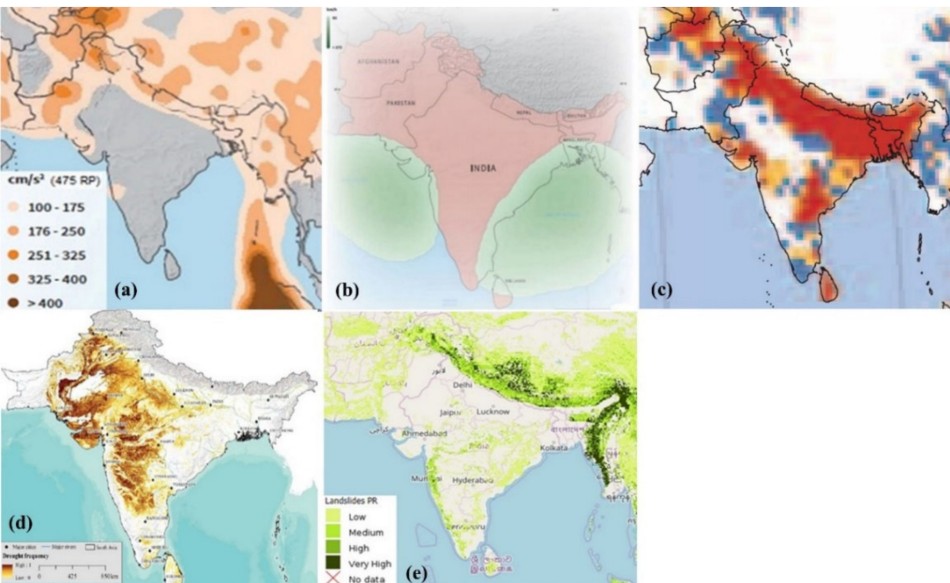

**Figure 1.** High-resolution exposure hazard maps of different DRMs: (**a**) Physical exposure to earthquakes, (**b**) physical exposure to cyclones, (**c**) physical exposure to riverine floods, (**d**) physical exposure to drought, and (**e**) physical exposure to landslides. (Sources: World Bank 2005 [37], UNEP/UNISDR 2013 [39], IASC/EC 2015 [42], UNISDR 2017 [15], IWMI 2017 [16]).

*3.2. Multi-Hazard Maps*

Multi-hazard maps (Figure 2) are useful for identifying locations with an elevated risk due to multiple hazards such as cyclones, droughts, floods, landslides and earthquakes. They show:

- Areas that are most likely to be exposed by multiple disasters;
- Disasters that occur immediately after the others, such as fires or landslides after earthquakes or cyclonic storms;
- Disasters that occur independently of each other with a considerable time lag, such as floods and droughts.

Disasters emanating from multiple hazards are frequent in South Asia. In the agricultural landscape, thousands of small tanks that are scattered everywhere provide relief from both floods and droughts to rural populations. However, extreme rainfall events and flash floods often damage many poorly maintained small tanks and reservoir storages, which reduces the resilience of communities against recurrent floods and droughts [49]. Urban centers have a high population density and infrastructure assets, and the industrial and service sectors there contribute substantially to economic growth. Floods and water scarcities associated with droughts are a substantial threat to the economic activities in urban centers.

Detailed multi-hazard maps are useful for the efficient planning of interventions. These are especially important for big countries such as India to identify the locations of multiple threats for risk management. Planning spatially targeted interventions for risk management is essential for the population facing numerous hazards to reduce exposure and vulnerability, or to improve the adaptive capacity. However, methodological differences contribute to the differences in risks between multi-hazard risk maps (Figure 2).

- WB 2005 and IWMI 2017 provide sub-national multi-hazard risk maps. WB 2005 has considered six hazards, while IWMI 2017 has considered five hazards. Only four hazards, tsunamis, floods, cyclones, and droughts, are common to both. Moreover,

in WB 2005, the sum of the individual risk values of the top three deciles (in 8th–10th deciles) is the multi-hazard index, where higher values indicate high-intensity multiple hazards. In IWMI 2017, a multi-hazard index (between 1 and 5) shows only the number of hazard occurrences in a pixel. Because some disasters have more prominence in the indicator value such as earthquakes in Nepal these two maps, to some extent, are not even comparable.

- IASC/EC 2015 and the University of Stuttgart 2015 show the aggregated risk of multi-hazards at the national level. They are thus adequate only for inter-country comparisons. They use a different number of hazards, indicators, and weights to aggregate indicators to assess risk components (exposure, vulnerability, and capacity). Moreover, the University of Stuttgart 2015 uses the arithmetic average, giving equal prominence to all risk components. However, IASC/EC utilizes the geometric average, giving differential prominence to different risk components. As a result, the risk pictures of some countries (Sri Lanka, Bhutan) show an opposite view.

Although there are differences, combining the information of multi-hazards and risks with local knowledge provides opportunities for designing efficient interventions to mitigate impacts (UNICEF 2014, WB/GFDRR 2011).

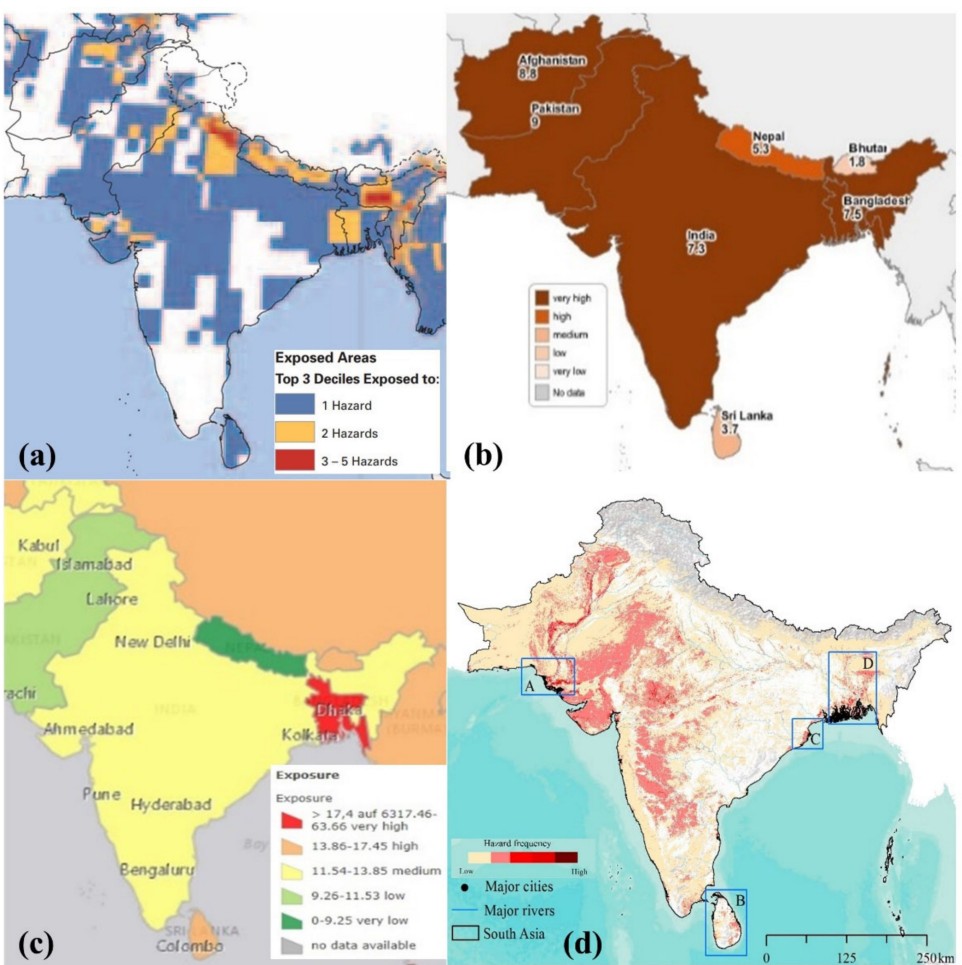

**Figure 2.** Physical exposure to multi-hazards. (Sources: (**a**) World Bank 2005 [37], (**b**) IASC/EC 2015 [42], (**c**) University of Stuttgart 2015 [43], (**d**) IWMI 2017 [16]).

**Table 4.** Details of data sources, spatial resolution, and intensity of hazard maps.

| Figures | | Assessment | Data Source | Pixel/Temporal Resolution | Return Period |
|---|---|---|---|---|---|
| Earthquakes | i | WB 2005 | Global Seismic Hazard Program (GSHAP) | 2.5′ | 475 |
| | ii | UNEP/UNISDR 2013 | UNEP/GRID-Geneva preview/GSHAP | 2.5′ | 475 |
| | iii | IASC/EC 2015 | GSHAP Seismic hazard map | 2.5′ | 475 |
| | iv | UNISDR 2017 | UNEP/UNISDR 2013 | 2.5′ | 475 |
| | v | University Stuttgart 2015 | UNEP/UNISDR 2013 | 2.5′ | 475 |
| | vi | EC 2017 | EMMI-GSHAP hazard map | 2.5′ | 475 |
| Cyclones | i | WB 2005 | UNEP/GRID-Geneva Preview | 30″ | 250 |
| | ii | UNEP/UNISDR 2013 | UNEP/GRID-Geneva preview | 30″ | 250 |
| | iii | IASC/EC 2015 | Annual physical exposure 1969–2009 | 30″ | 250 |
| | iv | UNISDR 2017 | UNEP/UNISDR 2013 | 30″ | 250 |
| | V | University Stuttgart 2015 | UNEP/UNISDR 2013 | 30″ | 250 |
| | vi | EC 2017 | UNEP/UNISDR 2013 | 30″ | 250 |
| Floods | i | WB 2005 | Dartmouth Flood Observatory, World Atlas of Large Flood Events | 1° | 200 |
| | ii | UNEP/UNISDR 2013 | UNEP/GRID-Geneva preview | 1° | 200 |
| | iii | IASC/EC 2015 | Annual physical exposure 1999–2007 | | |
| | iv | UNISDR 2017 | UNEP/UNISDR 2013 | | |
| | v | University Stuttgart 2015 | UNEP/UNISDR 2013 | | |
| | vi | EC 2017 | Flood map, JRC GloFAS | Raster/1 km | 100 |
| | vii | IWMI 2017 | MODIS | 500 m/8 days | - |
| Droughts | i | WB 2005 | IRI (International Research Institute for Climate Prediction Climate Data) Library | 2.5° | 8 days |
| | ii | UNEP/UNISDR 2013 | IRI Climate Data Library | | |
| | iii | IASC/EC 2015 | Not available | | |
| | iv | UNISDR 2017 | UNEP/UNISDR 2013 | | |
| | v | University Stuttgart 2015 | UNEP/UNISDR 2013 | | |
| | vi | IWMI 2017 | MODIS | 500 m/8 days | - |
| Landslides | i | WB 2005 | Not available | | |
| | ii | UNEP/UNISDR 2013 | Norwegian Geotechnical Institute (NGI) | 30″ | |
| | iii | IASC/EC 2017 | Not available | | |
| | iv | UNISDR 2017 | Not available | | |

Sources: WB 2005 [37], UNISDR 2015 [14], UNISDR 2017 [15], EC 2017 [44], IWMI 2017 [16] Notes: GSHAP—Global Seismic Hazard Program; IRI—International Research Institute for Climate Prediction; NGI—Norwegian Geotechnical Institute.

*3.3. Exposure Maps*

Exposure to hazards is hazard dependent. It is a function of the probability of the occurrence of hazards with different intensities and the elements of actual exposure to disasters that follow. Some of the common aspects exposed are the disaster-affected:

- Population (humans or livestock) and/or their mortality;
- Infrastructure or businesses and their value and output;
- Economic activities and their losses;
- Ecosystems and their services.

Assessment of the probability of hazard occurrence is dependent upon the availability of historical data, the period used for the analysis, and their spatial coverage. Therefore, exposure depends on the spatial coverage of selected hazards and exposed elements. Thus, multi-hazard exposure estimates of the population (Figure 3, Table 5) may provide different exposure profiles.

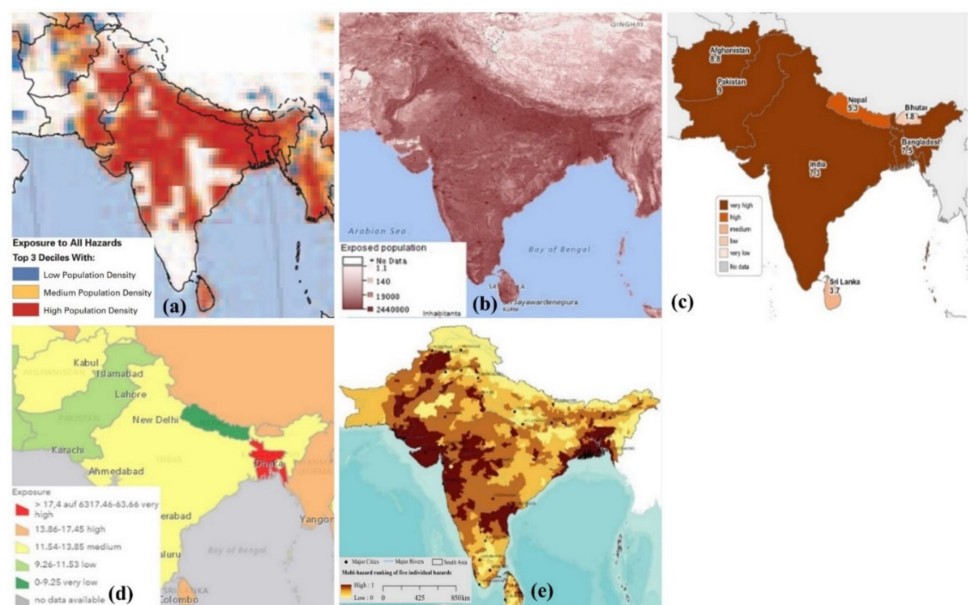

**Figure 3.** Population exposure to multi-hazards. (Sources: (**a**) World Bank 2005 [37], (**b**,**c**) IASC/EC 2015 [42], (**d**) University of Stuttgart 2015 [43], (**e**) IWMI 2017 [16]).

**Table 5.** Population exposed to disasters—% of the total population. (Sources: WB 2005 [37], IWMI 2017 [16], Pesaresi et al., 2017 [44]).

| Hazard | Assessment | % Population [1] Exposed to Disasters | | | | | |
|---|---|---|---|---|---|---|---|
| | | India | Pakistan | Bangladesh | Nepal | Sri Lanka | Bhutan |
| 3+ hazards [2] | WB 2005 | | | | | | |
| | IWMI 2017 | 1% | 2% | 2% | 0% | 1% | 0% |
| 2+ hazards [2] | WB 2005 | 11 | 18 | 33 | 51 | na | 29 |
| | IWMI 2017 | 11 | 15 | 20 | 2 | 12 | 14 |
| Floods | IWMI 2017 | 8 | 8 | 34 | 6 | 5 | 0 |
| | EC 2017 | 18 | 35 | 46 | na | na | na |
| Droughts | IWMI 2017 | 19 | 30 | 1 | 2 | 10 | 28 |

Notes: [1] Percentage relative to total populations in 2005 in WB 2005, and 2015 in IWMI 2017 and E 2017. [2] Only floods, droughts, extreme temperature are common to WB 2005 and IWMI 2017.

Regarding the exposure of the population exposure to multi-hazards:

- The IASC/EC 2015 and University of Stuttgart 2015 assess the exposure to similar hazards. The former shows high to very high exposure to multi-hazards for all countries except Sri Lanka and Bhutan, and for them, the exposure varies from medium to low. The University of Stuttgart 2015 depicts a substantially different exposure profile for the SA countries.
- The sub-national exposure estimates for multi-hazards in WB 2005, UNEP/UNISDR 2013, and IWMI 2017 also vary. The number and the intensity of hazard treatment in the risk analyzed area are also significant factors in these differences.
- The WB 2005 study shows that 33% of the population (in 2005) were exposed to three or more hazards in Bangladesh, and about 33%, 18%, and 11% of the total population in Bangladesh, Pakistan, and India, respectively were exposed to two or more hazards. A detailed higher spatial resolution risk study of Sri Lanka showed not only vast spatial variation but also a distinct seasonal variation of exposure to multi-hazard risks.
- The IWMI 2017 with a higher spatial resolution shows that two or more hazards affect about 20% of the population in Bangladesh and only 11% of the population in India. A substantially lower exposure in Nepal in IWMI 2017 was observed, due to the

non-inclusion of earthquakes in the assessment. IWMI 2017 study shows that about 750 million people in South Asia are affected by one or more natural hazards. Of these, 72% are in India, and 14% are in Bangladesh and Pakistan.

The WB 2005 reveals that natural hazards expose more than two-thirds of the GDP in SA countries (Figure 4). In the UNEP/UNISDR 2013, the GDP exposed to disasters ranges from USD 1.5 to 3 billion, which is substantially less than that estimated by WB 2005.

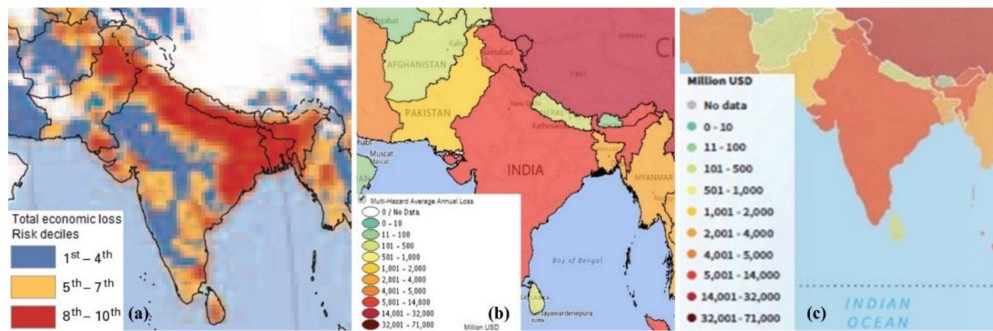

**Figure 4.** GDP exposed to multi-hazards. (Sources: (**a**) World Bank 2005 [37], (**b**) UNEP/UNISDR 2013 [39], (**c**) UNISDR 2017 [15]).

Despite these marked differences, global studies are still useful for prioritizing donor assistance for disaster risk reduction across countries. INFORM provides scientific support to a wide array of EU disaster risk assistance policy initiatives [50]. GRDP-UNISDR is the primary source of background information for policy dialogues at the biennial global gathering of member countries organized by the UNISDR. The exposure of the economy and GDP to multiple disasters is a valuable source for policymakers to take urgent action.

The individual disaster risk maps show the vast spatial spread of floods (Figure 5), droughts (Figure 6), and landslides (Figure 7) in most SA countries. However, the low spatial resolution in some studies (WB 2005, IASC/EC 2015) either masks or exposes more areas to floods and droughts. Table 6 shows that:

- IWMI 2017 estimated that floods affected 170 million people in SA. Of this, 101 and 53 million are in India and Bangladesh, respectively. However, according to EC 2017, the flood-affected populations in India and Bangladesh are 220 and 71 million, respectively.
- Estimates of the drought-affected population also vary. Figure 6 shows a different picture of drought exposure as depicted by the WB 20105 and UNEP/UNISDR 2013. IWMI 2017 estimated that droughts affected 293 million people in South Asia. Of this, 233 and 54 million are in India and Pakistan.

**Table 6.** Mortality and economic loss-related vulnerability coefficients of South Asia. (Source: World Bank 2005 [37]).

| Factor | Income Status | Cyclones | Droughts | Earthquakes | Floods | Landslides |
|---|---|---|---|---|---|---|
| Economic loss | Low | 26.64 | 0.18 | 1.33 | 7.00 | 0.07 |
| | Lower middle | 0 | 0 | 0 | 5.26 | 0 |
| | Upper middle | 0 | 0 | 0 | 0 | 0 |
| | Upper | 0 | 0 | 0 | 0 | 0 |
| Mortality | Low | 64.52 | 0.04 | 8.04 | 3.90 | 7.04 |
| | Lower middle | 0 | 0 | 0 | 0 | 0 |
| | Upper middle | 0 | 0 | 0 | 0 | 0 |
| | Upper | 0 | 0 | 0 | 0 | 0 |

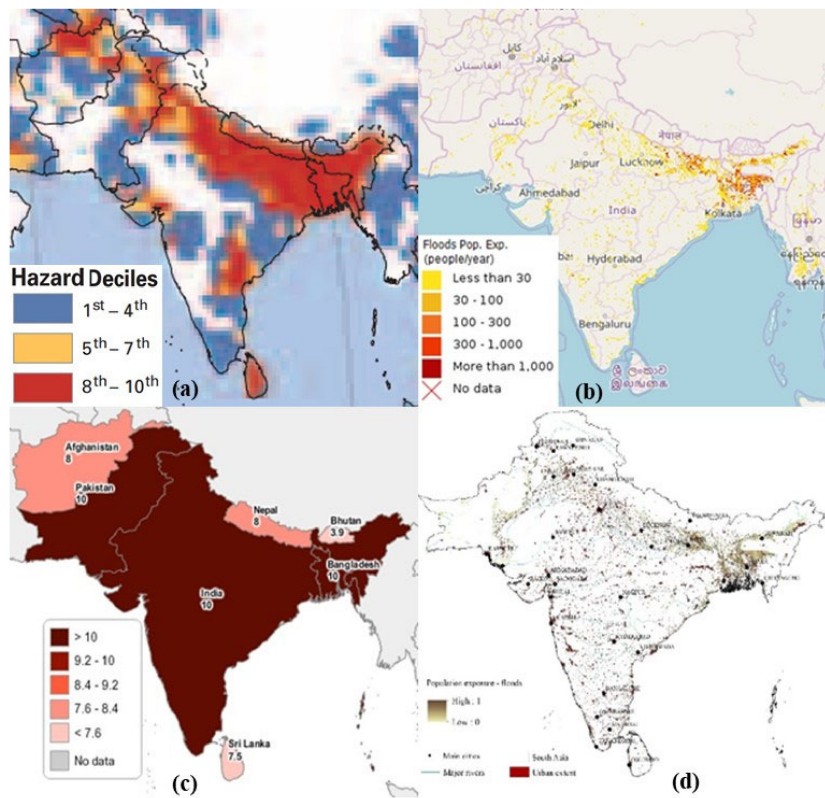

**Figure 5.** Population exposure to floods. (Sources: (**a**) World Bank 2005 [37], (**b**) UNEP/UNISDR 2013 [39], (**c**) IASC/EC 2015 [42], (**d**) IWMI 2017 [16]).

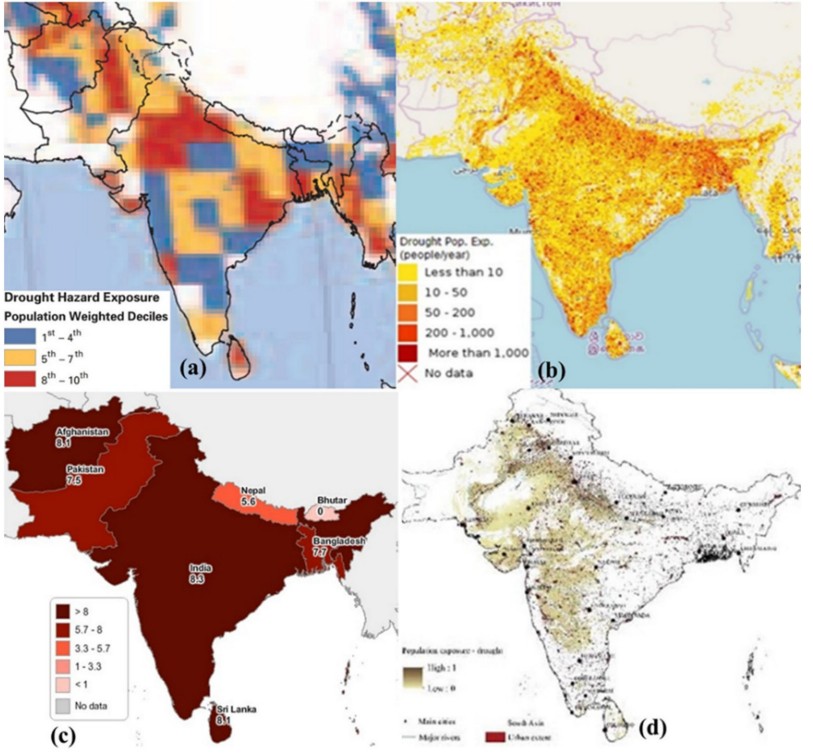

**Figure 6.** Population exposure to drought. (Sources: (**a**) World Bank 2005 [37], (**b**) UNEP/UNISDR 2013 [39], (**c**) IASC/EC 2015 [42], UNISDR 2017 [15], (**d**) IWMI 2017 [16]).

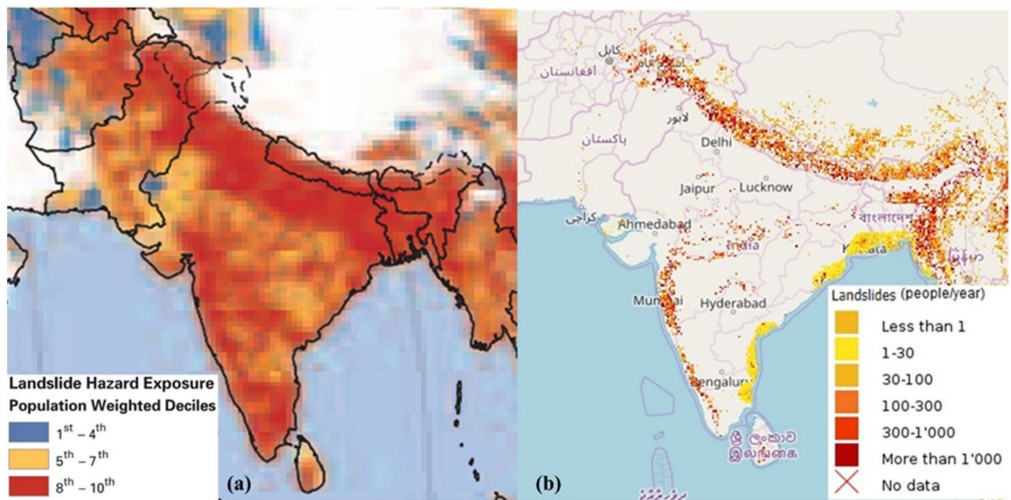

**Figure 7.** Population exposure to landslides. (**a**) World Bank 2005 [37], (**b**) UNEP/UNISDR 2013 [39].

*3.4. Vulnerability Maps*

Vulnerability includes both physical and social vulnerability. Often, physical vulnerability is hazard dependent. It assesses the vulnerability of infrastructure such as buildings, roads, bridges through vulnerability (or fragility) curves (World Bank 2015, UNISDR 2015). Social vulnerability is generally independent of hazards. It assesses the vulnerability of people, communities, and institutions by combining socio-economic, political, cultural indicators (IASC/EC 2015, University of Stuttgart, World Bank 2005, IWMI 2017). In addition to vulnerability, some studies assess coping capacity, which is independent of hazards. It measures the ability of people to cope with disasters. The latter includes two streams: institutional capacity and infrastructure capacity.

The comparison of assessments shows a lack of a coherent methodology underpinning the estimation of vulnerability. The vulnerability assessments of IASC/EC 2015 and the University of Stuttgart 2015 are deterministic, where the indices are from socio-economic, health, institutional, and infrastructure. However, there are substantial differences in the vulnerability estimates of these two assessments because they used different sub-indicators for the evaluation (Figure 8). On the other hand, IWMI 2017 used only HDI for the vulnerability assessment, where the HDI values are the vulnerability coefficients.

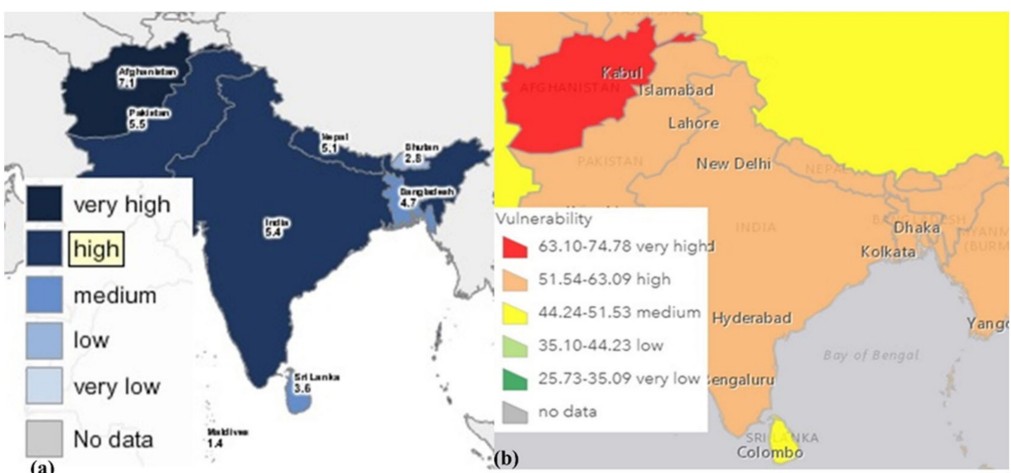

**Figure 8.** Vulnerability to multi-hazards. (**a**) IASC/EC 2015 [42], (**b**) University of Stuttgart 2015 [43].

The WB 2005, UNEP/UNISDR 2013, and UNISDR 2017 used a probabilistic approach using the historical data of mortality and economic losses to estimate vulnerability weights.

The WB 2005 estimated the vulnerability coefficients of mortality and losses using historical data for countries in the World Bank geographical region and income status. The estimated weights for the South Asia region in Table 6 are the economic loss per USD 100,000 of GDP and persons killed through 1981 and 2001 per 100,000 people in 2000. These estimates show that the low-income South Asian population has the highest vulnerability to death and economic losses due to cyclones.

The UNISDR-GRDP used historical data to assess vulnerability to hazards in different countries. This estimate includes vulnerability curves developed for each risk indicating mortality and potential losses concerning the intensity of hazards with varying periods of return. It is not possible to compare the vulnerability estimates of UNEP/UNISDR with those of IASC/EC 2015, University of Stuttgart 2015, and IWMI 2017, which used a deterministic approach through socio-economic details. Indirect comparison is possible through disaster risk maps, which are products of exposure and vulnerability.

### 3.5. Disaster Risks Maps

Two popular risk estimates are the risks to population and GDP. These are easy to estimate because the gridded population and GDP [51] estimates are available now. Still, there are substantial differences in calculated risks, especially the multi-hazard risks of various assessments. All assessments, except IWMI 2017, gave national level multi-hazard risks to GDP (Figure 9).

- WB 2005 shows high to medium risks for Sri Lanka, Bhutan, and Nepal, whereas UNEP/UNISDR 2013 shows low to substantially low risk for Sri Lanka, Nepal, and Bhutan. This variation may be due to the different spatial resolutions of hazard maps and different methods used for vulnerability estimation.
- The IASC/EC 2015 and University of Stuttgart 2015, which used a deterministic approach to vulnerability estimation, show a completely different risk picture. This difference may be due to various socio-economic development indicators and methods utilized for vulnerability estimation. For example, IASC/EC 2015 used a substantial number of socio-economic indicators for assessing vulnerability and coping capacity as compared to the University of Stuttgart 2015. Furthermore, the weights used for developing indexes are different. Moreover, IASC/EC 2015 used an arithmetic mean, while the University of Stuttgart 2015 used the geometric mean for estimating risk.

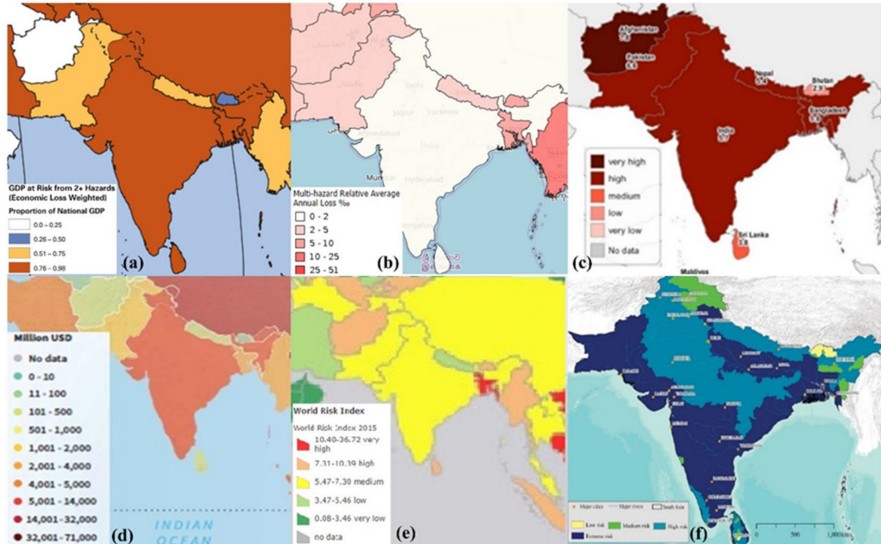

**Figure 9.** Spatial variation of multi-hazard risk. (**a**) World Bank 2005 [37], (**b**) UNEP/UNISDR 2013 [39], (**c**) University Stuttgart 2015 [43], (**d**) UNEP/UNISDR 2017 [15], (**e**) IASC/EC 2015 [42], (**f**) IWMI 2017 [16].

The sub-national multi-hazard disaster risk map of IWMI 2017 shows that a large swath of India has a very high risk of economic losses. This risk indicator, however, is different from others in that it used a sub-national HDI indicator, which included GDP, to assess risk. It provides useful information about sub-national risk, although it is not clear whether the exposure or vulnerability indicators dominated the risk assessment. Further analysis of the contribution of exposure and vulnerability to the final risk indicators could offer useful information to design appropriate risk management strategies.

*3.6. Social Development Indicators in Risk Maps*

The disaster risk maps show only a few social development indicators—mainly population, agriculture area, and GDP at risk. Many of the other social development indicators enter the risk analysis through the vulnerability component.

The selection of social development indicators shows a lack of a coherent methodology underpinning the estimation of vulnerability. The vulnerability assessments of IASC/EC 2015 and the University of Stuttgart 2015 are deterministic and used socio-economic indices on health, institution, and infrastructure. However, there are substantial differences in vulnerability values due to different sub-indicators used in the vulnerability assessments. On the other hand, IWMI 2017 combined HDI and hazard-affected populations for vulnerability assessment, where the HDI values are the vulnerability coefficients.

The gender inequality and age dimensions are only used in the risk assessments in IASC/EC 2015, ACTION Aid 2014, UNICEF 2014, and University of Stuttgart 2015. However, except in ACTION Aid 2014, gender-related indicators are subsumed under a host of other indices used for assessing vulnerability. Therefore, it is not clear how gender inequality and age dimensions influence the disaster risks of these two assessments.

Many national databases provide gender/age group-specific data such as population and employment at the sub-national level. Such estimates can provide gender and age group-specific exposure to hazards. The probabilistic assessment of gender and age-group vulnerability is still not possible due to a lack of information on gender/age group-specific mortality or economic losses. However, the method employed by IWMI 2017, which used HDI to assess vulnerability, can overcome this deficiency. By combining HDI with other social development indicators that represent gender-related issues, the vulnerability assessments of gender can be ameliorated. The Women Resilient Index of Action Aid 2014 incorporated many such gender-related matters.

The WRI-EIU used 68 indicators (Table A2) on social, economic, institutional, and infrastructure aspects to analyze the ability of South Asian countries with respect to disaster risk reduction (DRR) and women's roles in DRR [52]. Of these, 40% are gender-disaggregated indicators. Hazard exposure and vulnerability assessments can use those gender-related indicators where data are available at the sub-national level. Some of these include gender-disaggregated data on

- Access to financial institutions;
- Access to micro-finance;
- Access to loans;
- Access to land;
- Unemployment rate;
- The number employed in the police force;
- Enrolment in primary-secondary schooling;
- Literacy rate, etc.

## 4. Discussion

*4.1. An Example of Disaster Risk Assessments from the Asian Development Bank's (ADB) Risk Strategies*

The ADB proposes a disaster risk assessment (DRA) for countries with medium- to high-risk [53]. The high-risk countries are those with absolute Average Annual Losses (AAL) due to multi-hazards above 2% of the total GDP. The medium-risk countries are

considered to be those with AAL between 0.8% and 2.0% of GDP. However, the DRAs should include low-risk countries with a large geographical area—such as India, which has pockets of high-risk regions—or those prone to hazards such as drought or insignificant earthquake zones, where the impacts of drought are often not included in AAL estimation. Earthquakes have long return periods, but when they do occur, they cause substantial damage.

Among the developing member countries (DMCs), high-risk members are often low- to middle-income countries [54]. In South Asia, Bangladesh and Bhutan are in the high-risk category with AALs of more than 2% of the GDP. India has the third largest AAL in Asia and the Pacific, but it is only 0.5% of the total GDP. All other DMCs in South Asia, except Sri Lanka and the Maldives, have a medium risk, where the expected AAL is between 0.8% and 2.0% of the total GDP. With a significant spatial and temporal variation in climate, large countries such as India or even island nations such as Sri Lanka can also have areas with significantly higher risks to disasters.

Because of the increasing incidence and increasing losses, the ADB proposes that DRAs should be part of the development policy and planning within countries [55]. The DRAs not only help in the planning of sustainable development projects, they also assist in enhancing resilience against disasters. Those, in turn, contribute to achieving sustainable development goals. Therefore, to discuss DRA and risk management strategies along with development projects with the relevant government officials, ADB proposes to include DRA as part of the country partnership strategies (CPS). The CPS is the springboard for the ADB to initiate discussions with the local governments on disaster risk management in development assistance.

In developing projects, the ADB project teams conduct a preliminary climate risk screening. They use a variety of methods, including the analysis of secondary data, a review of the published documents, and an assessment of risks using the online tool AWARE for such projects. Preliminary climate risk screening with AWARE was utilized in the Earthquake Emergency Assistance Project in Nepal, Thimpu Road development project in Bhutan, Dhaka Water Supply Network Improvement Project in Bangladesh, etc. While the preliminary risk screening of the ADB generally has a low spatial resolution (e.g., districts of Nepal in Figure 10), in project locations in high-risk areas, detailed localized risk assessments are conducted.

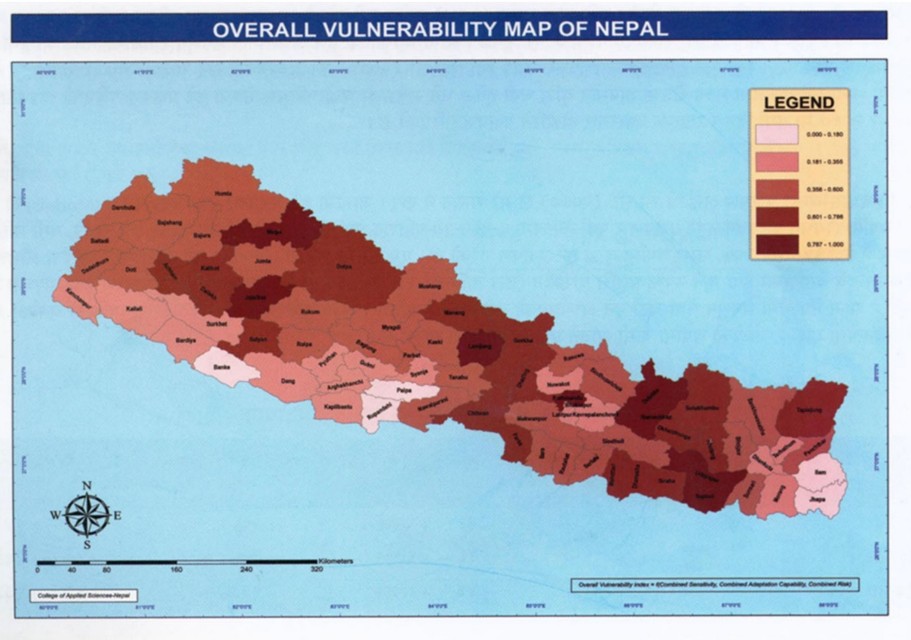

**Figure 10.** AWARE generated risk map.

*4.2. Modeling and Future Applications*

It is evident from the desk review that in order to manage natural disasters, mapping hazards and identifying areas for risk prioritization and planning are crucial. Most natural hazards studies as of now have focused on single hazards. However, there should be greater emphasis on the relationships between multiple hazards to collectively quantify risks and their impacts for medium- to long-term resilience strategies. Given the complexity of hazards and their exposure and vulnerability indicators, it is vital to integrate a machine learning framework for multi-hazard modeling and mapping to help understand the complex relationships. Various methods exist—namely, support vector machine (SVM), analytical hierarchy process (AHP), decision trees, and multivariate statistical analysis to combine hazards with the use of multiple variables with statistical weighing—for mapping multi-hazards risks. In addition to modeling, there is a need for expert opinion, the use of remote sensing information, and policy consultation to provide stakeholders with robust information for proactive climate adaptation strategies.

Innovations created through technological advancement are more effective in assisting disaster risk management, resilience, and response processes, as well as providing new dimensions for data analysis. Innovations in areas such as Artificial Intelligence (AI), Internet of Things (IoT), 4G-5G wireless network, and Machine Learning (ML) have overturned pre-existing methods in many areas, including disaster risk reduction and management. Furthermore, the past few decades have seen significant technological advances in areas such as remote sensing and information technology, resulting in the increasing use of satellite and drone data, smartphones, and social media, as well as the Internet. Moreover, technological advancements, as well as the use of data and tools, has led to a proliferation of Big Data platforms that provide additional information on disaster risks and impact. IoT, Big Data, ML, and AI are currently at the forefront of digital transformation around the world and will play a significant role in the development of comprehensive disaster risk management and resilience strategies with a high degree of efficiency [56].

Disaster/risk models are currently being used for a variety of stages throughout the DRM cycle, making them more useful for infrastructure planning, insurance products, and early warnings. Such risk models are primarily designed by integrating hazards, vulnerability, exposure elements, risk indicators, and historical impact data. As the amount of data retrieved increases day by day, there is a high potential to change the way disaster risk modeling and management is undertaken by combining it with artificial intelligence (AI). AI uses include real-time analysis of seismic data for forecasting and detection models, the identification of data communication patterns through social media in the event of disasters, and the generation of flood forecasting models.

## 5. Conclusions

Global DRMs mainly assess national-level disaster risks and provide broader climate adaptation strategies. The core of all global DRMs includes the assessment of single or multiple hazards, exposure, and vulnerability. However, the risk profiles of countries are mostly not comparable across assessments due to inconsistent approaches and methodologies used for the quantification of exposure, susceptibility, and risks. Moreover, many of the global DRMs do not provide detailed sub-national maps, which are critical for local-level development and disaster risk management planning. Therefore, regional studies with sub-national entities acting as the analytical units to support data sharing and evaluation with multi-institutions might be the best way forward in the generation of better disaster risk maps for South Asia.

At present, the gridded risk is mainly available for population and GDP. However, detailed production data on agriculture, water scarcity or security, and other sectors are available at the district level in South Asia in national/province/state-level databases. These databases also compile many other socio-economic development indicators. These can generate sub-national administrative boundary level risk estimates for population, agri-

culture and different sectoral outputs, and GDP. These need to be extended to other social development and demographic indicators, including gender and other vulnerable groups.

Risk assessments at the gender and age dimensions are rare. However, in some global DRMs, gender inequality and age dimensions are combined in risk estimation through a vulnerability assessment. However, their influence on risk estimates is not apparent due to the large number of indicators that are used for vulnerability assessments. Yet, there is potential to estimate gender and age division risk profiles by using sub-national data collected by the various census. A deterministic vulnerability assessment can use detailed census data to assess risks at a sub-national or project level aided with a machine learning framework to produce reliable multi-hazard risk maps. It is important to identify a set of consistent indicators where data are available from the population and another census, to use in vulnerability and risk assessments [57].

In this respect, establishing a robust evidence base through the collection of data on current and projections of future hazards and disasters is crucial to disaster risk assessment, financing, and management. The information base requires (sub)national, regional, and global databases capable of pooling data from diverse sources. It enriches risk assessment and enables the development of more cost-effective, innovative disaster risk financing tools and insurance products such as Index-Based Flood Insurance [58] and the Parametric product for Storm, Cyclone or Hailstorm [59].

The use of remote sensing and GIS to assess and integrate hazard, exposure, and risk indicators with higher spatial resolutions to sub-national risk maps is preferable because existing global DRMs do not provide a sectoral-wide risk. These estimates are essential given the increasing influence of disaster risk on industrial and service sectors involved in economic growth. In order to improve the accuracy and rapid mapping of multi-hazard risks, it is important to use advance models such as the machine learning approach to guide policymakers for timely hazard mitigation measures. Such assessments allow the policymakers, development practitioners, and the private sector including the insurance industry to manage current and future risks through risk reduction, risk transfer, and risk management instruments. High-precision multi-hazard risk maps should also be part of the global, regional, national, and sub-national implementation plans to strengthen the Sustainable Development Goals (SDG), the Paris Climate Change Agreement, and the Sendai Framework for Disaster Risk Reduction towards building resilience among vulnerable communities in South Asia.

**Author Contributions:** G.A. and U.A.A. conceptualized the study. U.A.A., G.A. and N.A. performed the review and written the paper. All authors have read and agreed to the published version of the manuscript.

**Funding:** This research was funded by the CGIARs (Consultative Group of International Agricultural Research Program (CRP) on Climate Change, Agriculture and Food Security (CCAFS), Asian Development Bank (ADB) and CGIAR Research Program (CRP) on Water, Land and Ecosystems (WLE).

**Institutional Review Board Statement:** Not applicable.

**Informed Consent Statement:** Not applicable.

**Data Availability Statement:** Not applicable.

**Acknowledgments:** The authors would like to acknowledge and thank the funding and data-providing organizations. We are grateful to three anonymous reviewers and the editors for their comments, which greatly improved the manuscript.

**Conflicts of Interest:** The authors declare no conflict of interest.

# Appendix A

**Table A1.** List of various disaster risk mapping tools and their application.

| No. | Name and Abbreviation | Agency and Year (Updates If Available) | Objectives (Link to Risk Maps or Mapping Tools) |
|-----|------------------------|------------------------------------------|--------------------------------------------------|
| 1 | Natural Disaster Hotspot: A Global Risk Analysis (Hotspot Study) | WB 2005 | The project enhances the knowledge of global risks of natural disasters by conducting an assessment with global datasets but taking into account the spatial variation of hazards, exposure, and vulnerabilities of six natural hazards. http://documents.worldbank.org/curated/en/621711468175150317/pdf/344230PAPER0Na101official0use0only1.pdf (accessed on 18 April 2021) |
| 2 | Open Data for Resilience Initiative (Open DRI) | WB/GFDRR 2011 | OpenDRI provides information about the rapidly changing dynamics of the risks and impacts due to population growth, economic expansion, and climate change. OpenDRI promotes spatially targeted investments, policy, or technical interventions that enhance the resilience of people and communities TABLEagainst climate change impacts. https://opendri.org/ (accessed on 18 April 2021) |
| 3 | Global Risk Data Platform (GRDP) | UNEP/UNISDR 2013 | The UNISDR organizes a biennial global gathering of member countries on reducing disaster risk and enhancing the resilience of communities and nations. The *Global Assessment Reports* (*GARs*) are UNISDRs flagship publications of Global Platform meetings. The GRDP is the warehouse of spatial information of exposure and risks generated by stakeholders for the biennial gatherings. http://preview.grid.unep.ch/index.php?preview=home&lang=eng (accessed on 18 April 2021) |
| 4 | Child-centered Risk Assessment: Regional Synthesis of UNICEF Assessments in Asia | UNICEF 2014 | UNICEF, with a mandate for humanitarian relief and development, especially for children, promotes child-centered disaster risk assessment. It informs the governments, the UNICEF country offices, and partner organizations on assessments of disaster risks for the survival and development of children. The evaluation explores ways to reduce vulnerability and build capacity to enhance resilience against disaster risks. https://www.preventionweb.net/publications/view/36688 (accessed on 12 April 2021) |
| 5 | South Asia Women's Resilience Index (WRI) | Action Aid 2014 | The WRI-EIU shows the extent to which the disaster risk reduction and building national resilience initiatives in the South Asian counties incorporated gender inequality in risk estimation. http://www.actionaid.org/australia/digital-tool-womens-resilience-index-wri (accessed on 12 May 2021) |
| 6 | Index for Risk Management (INFORM) | IASC/EC 2015 | INFORM's objective is to assess countries that are at a potentially high risk of hazards and inform the world and donors for prioritizing for international humanitarian assistance. http://www.inform-index.org/ (accessed on 14 Apr 2021) |
| 7 | The World Risk Index (WRI) | University Stuttgart 2015 | The World Risk Index combines physical hazard information with vulnerability (susceptibility, coping, and adaptive capacity) to assess the risk of people exposed to disasters. It provides the likelihood of natural hazards affecting people and their vulnerability to hazards. http://www.uni-stuttgart.de/ireus/Internationales/WorldRiskIndex/ (accessed on 11 June 2021) |
| 8 | The GAR Atlas: Unveiling Global Disaster Risk (GAR Atlas) | UNISDR 2017 | The GAR Atlas presents the results of the Global Risk Model, which uses a state-of-the-art probabilistic approach to assess hazards, exposure, and vulnerability. The UNISDRs GAR report [15] previewed the initial results of the Global Risk Model. https://www.preventionweb.net/english/hyogo/gar/atlas/ https://www.unisdr.org/we/inform/publications/42809 (accessed on 11 April 2021) |
| 9 | Atlas of the Human Planet: Global Exposure to Natural Hazards | EC 2017 | The Atlas shows the spatial patterns and temporal trends of exposure of human settlements to disaster risk and their relation to socio-economic vulnerability. They draw attention to geographical hotspots for a comprehensive understanding of disaster risks. https://ec.europa.eu/jrc/en/publication/eur-scientific-and-technical-research-reports/atlas-human-planet-2017-global-exposure-natural-hazards (accessed on 19 May 2021) |
| 10 | Mapping Multiple Climate-related Hazards in South Asia | IWMI 2017 | This study develops a high spatial resolution mapping of areas exposed to multi-climatic hazards and estimates exposure and vulnerability of population and agriculture to multi-hazard disaster risks. http://www.iwmi.cgiar.org/publications/iwmi-research-reports/iwmi-research-report-170/ (accessed on 28 May 2021) |

## Appendix B

**Table A2.** Indicators used for vulnerability assessment.

| | Indicators in Different Risk Assessments | WB 2005 | UNEP/UNISDR 2013 | WB/GFDRR 2011 | UNICEF 2014 | ACTION Aid 2014 | IASC/EC 2015 | University of Stuttgart 2015 | EC 2017 | UNISDR 2017 | IWMI 2017 |
|---|---|---|---|---|---|---|---|---|---|---|---|
| 1 | GDP per capita | x | | | x | | | | | | |
| 2 | Human development index | | | | | | x | | x | | x |
| 3 | Human poverty index | | | | x | | x | | | | |
| 4 | Extreme poverty | | | | | | | | | | |
| 5 | GINI index | | | | | | x | | | | |
| 6 | Adult literacy rates | | | | | x | x | x | | | |
| 7 | Gender inequality index | | | | | | x | x | | | |
| 8 | Public aid received (total and % of GDP) | | | | x | | x | | | | |
| 9 | Displaced people (total and % of the total population) | | | | | x | x | | | | |
| 10 | Dependency ratio | | | | | x | x | | | | |
| 11 | Prevalence of TBC | | | | | | x | | | | |
| 12 | Malaria mortality | | | | | | x | | | | |
| 14 | Adult HIV cases | | | | | | x | | | | |
| 15 | Children underweight | | | | | | x | | | | |
| 16 | Children morality rate | | | | | | x | | | | |
| 17 | Health expenditure per capita | | | | | | x | x | | | |
| 18 | 1-year old fully immunized against | | | | | | x | | | | |
| 19 | Prevalence of undernourishment | | | | | | x | | | | |
| 20 | Average dietary energy supply adequacy | | | | | | x | | | | |
| 21 | Life expectancy at birth | | | | | | | x | | | |
| 22 | The domestic food price index | | | | | | x | | | | |
| 24 | Domestic food price volatility index | | | | | | x | | | | |
| 25 | Number of physicians per 10,000 inhabitants | | | | | x | x | x | | | |
| 26 | Governance ineffective index | | | | | | x | x | | | |
| 27 | Corruption perceptions index | | | | | | x | x | | | |
| 28 | Access to electricity | | | | x | x | x | | | | |
| 29 | Number of women in the police force | | | | | x | | | | | |
| 30 | Internet users | | | | | | x | | | | |
| 31 | Mobile phone subscriptions | | | | | | x | | | | |
| 32 | Road density | | | | | | x | | | | |
| 33 | Access to improved water supply | | | | | x | x | | | | |
| 34 | Access to clean sanitation | | | | | x | x | | | | |
| 35 | Number of hospital beds 10,000 inhabitants | | | | | x | | x | | | |
| 36 | Insurance availability | | | | | | | x | | | |
| 37 | Share of females in the national parliament | | | | | | | x | | | |
| 38 | Water resources (Envir. security index) | | | | | | | x | | | |
| 39 | Biodiversity and habitat productions | | | | | | | x | | | |
| 40 | Forest management | | | | | | | x | | | |
| 41 | Agriculture management | | | | | | | x | | | |
| 42 | Government funding for disaster relief | | | | | x | | | | | |
| 43 | Country-level economic strength | | | | | x | | | | | |
| 44 | Personal finance of women | | | | | x | | | | | |
| 45 | Labor environment of women | | | | | x | | | | | |
| 46 | Communication | | | | | x | | | | | |
| 47 | Quality of power supply | | | | | x | | | | | |
| 48 | Environmental sanitation | | | | | x | | | | | |

Source: World Bank 2005 [37].

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
