# Peer review of "Disaster Risk Mapping: A Desk Review of Global Best Practices and Evidence for South Asia"

_sustainability, doi:10.3390/su132212779_

Round 1

Reviewer 1 Report

The paper summarizes well the literature and available data and mapping techniques for disaster risk mapping. The paper is not a traditional research paper but it does make a contribution to the literature. Hence, the paper would benefit by establishing unambiguous objectives in the introduction, describing in detail what aspects of the literature it will cover. The abstract includes clear objectives that those could also be described better in the introduction. 

Author Response

Reviewer comments attached.

Reviewer 2 Report

A great paper!

Please find below some suggestions to improve your paper:

1. Present research questions in the introduction

2. Reasons the author selected these disaster risk tools and assessments to review (why not others? Why only 10) (added to line 92, 93). One of the framework that might be deserved to be included is National Disaster Risk Assessment – Guidelines: Governance System, Methodologies, and Use of Results of UNISDR in support of the Sendai Framework for DRR 2015 – 2030 (https://www.unisdr.org/files/52828_nationaldisasterriskassessmentpart1.pdf)

3. Discussion 4.1 on Social Development Indicators in Risk Maps should be one of findings more than a part of discussion.

4. Discussion 4.2 on Disaster Risk Assessments Example from Asian Development Bank (ADB) Risk Strategies should be considered to add some more information: why did you choose it as an example and example of what, the strength and weaknesses and application in practices.

Author Response

Reviewer - 2 comments attached here.

Reviewer 3 Report

  1. The introduction part is detailed and well written. The need for a detailed disaster risk assessment is provided with data supporting from Bank’s Aftershocks report with increasing incidences of natural hazards.
  2. The incidences of natural hazards are not mentioned clearly. Floods affect on India and Sri Lanka and some of the incidences has been listed but it affects other south Asian countries also.
  3. Year wise data can also be provided to give an account of occurrence of different natural hazards.
  4. The disaster risk assessments used are till 2017 only may be enhanced to record recent disasters impacts.

RESULTS

Individual Hazard Maps

  1. The national-level hazard maps are sufficient for comparisons across countries.
  2. Higher resolution hazard maps show considerable details of hazard exposure.
  3. IWMI 2017 shows the actual exposure.
  4. Figure 1 has different country boundaries, and (a) and (c) map is tilted.
  5. Boundary for India is not standard as per SOI maps, hence note may be placed about inaccuracy in country boundaries as available in diffrent sources.

Multi-Hazard maps

  1. Figure 2 represented in (a) exposed areas for multiple hazards which has not mentioned which hazards. As which areas are particularly exposed by different disasters and which color in the image represents what.
  2. In the same figure (c) the image is stretched horizontally.
  3. The figure 2(d) is not clear and the legend is not visible.

Exposure maps

  1. Figure 3 is clear and easy to understand but the maps are aligned differently, the a and c are vertically stretched and a is also tilted, b seems and d seems good, the legend is not clear in e.
  2. Same concern in figure 4 also.

Vulnerability Maps

  1. Vulnerability maps and data are well explained.

Disaster risk maps

  1. WB 2005 shows high to medium risk for Sri Lanka, Bhutan and Nepal whereas UNEP/UNISDR 2013 shows low to substantially low risk for Sri Lanka, Nepal and Bhutan.
  2. The IASC/EC 2015 and University of Stuttgart 2015, which used a deterministic approach to vulnerability estimation, show a completely different risk picture, need to be discussed

Modelling and future application

  1. It is vital to integrate a machine learning framework for multi-hazards modelling and mapping to help understand the complex relationships. Various methods exist — namely support vector machine (SVM), analytical hierarchy process (AHP), decision trees, and multivariate statistical analysis to combine hazards with the use of multiple variables with statistical weighing — for mapping multi-hazards risks as mentioned in review.
  2. More focus and details can be provided on use of remote sensing for disaster risk mapping as it is the best possible way for assessing risk assessment for smaller areas (on large scale basis) as compared to the national level assessments available.

CONCLUSION

  1. The risk profiles are not comparable due to inconsistent approaches and methodologies as mentioned in the review. more clarity be be given.

Author Response

Reviewer-3 comments responded and available in the attached document.

Round 2

Reviewer 1 Report

The paper makes good contributions.

Reviewer 3 Report

 revision done as suggested